# The Relationship between Locomotive Syndrome Risk, Gait Pattern, and Standing Posture in Young Japanese Women: A Cross-Sectional Study

**DOI:** 10.3390/healthcare8040565

**Published:** 2020-12-15

**Authors:** Yuichi Uesugi, Saki Kanaya, Hiroko Nakanishi, Yoshihiko Naito

**Affiliations:** 1Department of Food Sciences and Nutrition, School of Food Sciences and Nutrition, Mukogawa Women’s University, Hyogo 663-8558, Japan; naito@mukogawa-u.ac.jp; 2Ashiya Municipal Asahigaoka Elementary School, Hyogo 659-0012, Japan; kanaya.103@edu-ashiya.ed.jp; 3Department of Innovative Food Sciences, School of Food Sciences and Nutrition, Mukogawa Women’s University, Hyogo 663-8558, Japan; n_hiroko@mukogawa-u.ac.jp

**Keywords:** locomotive syndrome risk, walking speed, posture, surveys and questionnaires, young women, Japan

## Abstract

Young people are also at risk of developing locomotive syndrome for unclear reasons. Therefore, we sought to evaluate the locomotive syndrome risk in young Japanese women and the relationship between standing posture and gait patterns. We used survey materials for physical measurements, locomotive syndrome risk tests, normal and maximum walking test, a standing posture test, and physical activity measures. A questionnaire-based cross-sectional survey was conducted with 100 Japanese female university students. The participants were divided into two groups (high-risk and low-risk groups) based on locomotive syndrome risk tests. The high-risk group accounted for 65.0% of the total participants. The high-risk group had a significantly slower walking speed and lower walking stride length than the low-risk group during maximum walking. Additionally, this high-risk group had a more prone posture than the low-risk group. Furthermore, the low-risk group included more individuals who belonged to middle and high school athletic clubs than the high-risk group. The locomotive syndrome risk was related to the walking pattern, standing posture, and past exercise habits. Therefore, long stride length, correct standing posture, and exercise habits acquired from a young age are important measures for preventing locomotive syndrome in young adults.

## 1. Introduction

The proportion of people certified for long-term care needs is increasing in Japan. The risk factors for this significant increase are related to locomotive syndromes, such as poor gait function and posture deterioration [1]. Deterioration in walking function due to aging reduces walking speed; reduction in stride length is more important than reducing cadence [2,3]. Factors that decrease stride length in the elderly include prolongation of both legs’ supporting period due to deterioration in balance ability, reduced lower-limb muscle strength, decrease in leg elevation during the swing phase, increase in step intervals, and decrease in arm swing range of motion [3,4]. These factors render walking unstable and lead to long-term care linked to falls and fractures [5]. Furthermore, the sway of the center of gravity increases because of deterioration in standing postures and walking function deteriorates [6,7]. Therefore, stabilizing the standing posture and preventing impaired walking function lead to preventive care [6,7]. However, young women have problems with “thinness” accompanied by muscle loss and atrophy, and “hidden obesity” with normal BMI and low lean mass [8,9]. The presence of muscle atrophy (sarcopenia) has been reported in women in their 20s. It has also been reported that the lower limb muscle strength of elementary school students is significantly reduced [10,11]. In a study of first-year students at a women’s college, Uesugi et al. reported that 26.6% of the participants were at risk for locomotive syndromes associated with body composition and lifestyle [12]. As women age, they lose muscle mass and strength, especially in the lower limbs [13]. In particular, the peak cross-sectional area of the quadriceps is in their twenties [14]. The female bone mass has been reported to be at a peak in their twenties [15]. With these declines, it is expected that motor function will decline, and the risk of fractures and falls will increase. In other words, it is important to take measures from a young age to prevent these problems. Therefore, some young women are at locomotive syndrome risk, and if this is overlooked, they might need long-term care for locomotive syndrome in the future. In addition, women at locomotive syndrome risk may present potential clinical risks such as poor physical fitness, poor walking, and poor standing similar to the elderly. However, to the best of our knowledge, no studies have been conducted to confirm the relationship between locomotive syndrome risk and physical fitness, gait index, and standing posture in young people. Therefore, this study aimed to evaluate the relationship between the risk of locomotive syndrome in young Japanese women and the standing posture and walking pattern.

## 2. Materials and Methods

### 2.1. Survey Target and Period

The participants of this study were recruited from a women’s university in Japan. The recruitment method included posters and electronic bulletin boards on campus for two months. The recruitment criteria included all undergraduate students at this university and those without problems with walking function. Overall, 103 students applied and they were all measured. In this analysis, the participants were 100 (97.1%) individuals aged 18–23 years (20.0 ± 1.5 years), excluding one who refused to body mass and two who were older than 23 years. We explained that participation in the survey for this study was voluntary and that consent could be withdrawn at any time without any disadvantage. All measurements were obtained in October 2015. Participants completed all measurements in one day.

### 2.2. Measurement and Measuring Equipment

#### 2.2.1. Physical Measurement

Physical measurements included height, body mass, body composition, and calf circumference. Height was measured using a height gauge (YS101-S, Yoshida Seisakusho Co., Ltd., Nishinomiya, Japan). Body mass and body composition were measured using a body component analyzer (InBody430, InBody Japan Inc., Tokyo, Japan). Body composition was estimated by skeletal muscle mass, body fat mass, and body fat percentage using the bioelectrical impedance method. The body mass index was calculated by dividing the body mass by the square of height. Regarding the lower limb circumference, the knee joint was bent at 90 degrees in a sitting position, and the circumference of the thickest part of each calf was measured with a measuring tape. Then, the left and right average values were calculated [16].

#### 2.2.2. Locomotive Syndrome Risk Tests

The locomotive syndrome risk test includes the “stand-up test”, “two-step test”, and “the 25-question geriatric locomotive function scale” advocated by the Japanese Orthopedic Association [17,18]. In this study, we performed the “stand-up test” and “two-step test”. The stand-up test used four seats with different heights of 40 cm, 30 cm, 20 cm, and 10 cm. First, the participants stood up from a 40 cm seat with one foot and maintained their posture for 3 s. Participants also stood on the other leg. If they could complete the task on both legs, they lowered the seats by 10 cm and repeated the same procedure. The evaluation value was based on the height of the lowest platform from which the participant could stand on both legs. The two-step test measured two strides when two steps (duplicate stride) were taken with maximum stride length. At that time, we ensured that either of the participants’ feet was on the floor. We also paid sufficient attention to their falls. We calculated their two-step values by dividing the length (cm) by their height (cm). They performed both tests twice, and we selected the best result.

#### 2.2.3. Walking Tests

We adopted the 10-m Walking Test [19]. Walking tests were performed on a 16-m straight walkway on a flat floor. We measured the number of steps and time of the 10 m section except for the 0–3 m and 13–16 m sections. The participants performed normal walking and maximum walking three times each. We calculated “the walking speed (m/s)” and “the average stride length (cm)” using the average value of normal walking and the maximum value of maximum walking.

#### 2.2.4. Standing Posture Test

We attached markers to participants in 21 locations (ears, distal ends of metacarpal III, distal ends of phalanges, shoulder joints, elbow joints, hand joints, hip joints, knee joints, foot joints, vertex, upper margin of sternum, and navel). Furthermore, we asked the participants to assume an erect and immobile posture for photographs and to stand at attention in front of a wall with horizontal lines drawn at 10-cm intervals. We installed a digital camera (D3100, Nikon Co., Tokyo, Japan) at the height of 80 cm, 1 m away from the participants, and acquired still participants’ images from the coronal and sagittal planes. We performed a two-dimensional image analysis from the image of the sagittal plane [20,21]. We printed out an enlarged photo of each participant, and we drew a line on each segment from the marker. Further, we calculated the position of gravity of each segment obtained with reference to the mass ratio of each part and position of the center of gravity of the Japanese body [20]. Then, we calculated the body center of gravity (COG) from the mass ratio. We drew a midline from the COG and measured the distance between the line and ear position. The distance was calculated as the “degree of forward lean (cm)” of the head (Figure 1) [21].

#### 2.2.5. Motor Function Tests

Three motor function tests were conducted: leg extension, closed eye one-leg standing, and functional reach tests. The lower extremity extension power test was performed using a leg extension power measuring device (Anaero Press 3500, Combi Co., Tokyo, Japan). The participants sat upright in the equipment’s chair, with both knees in a 90-degree flexion position, extended the leg at the knee as quickly as possible after 5 counts, and pushed the footplate at their feet. The measurement was performed 5 times, and the measurement interval was 15 s. We selected the highest leg extension value out of the five measures. In the one-leg standing balance test with eyes closed, the participants were standing on one leg with both eyes closed, and the time required for the raised foot to reach the floor was measured. The test was terminated when the axial foot left the floor, was significantly displaced, or the raised leg touched the axial foot. The test was repeated twice, and the result with the higher value was selected. The maximum time for one test was 180 s. The fist was closed in the functional reach test, and one shoulder was bent at 90 degrees while standing. We instructed the participants to “reach forward as far as you can without taking a step.” The moving distance of the fist between the starting and finishing positions was measured. The test was successfully performed twice. The one with the greater distance value was adopted as the measurement result.

#### 2.2.6. Questionnaire Survey

The questionnaire survey was a modification of the Japan Arteriosclerosis Longitudinal Study Physical Activity Questionnaire for students (JALSPAQ) [22,23]. The JALSPAQ was modified to be suitable for student life. Specifically, we changed the wording from “work” to “school life” and included items such as part-time jobs in leisure activities. We have previously validated this change for college students. We asked about the frequency of daily and weekly sleep, school life, mobility, housework, and leisure activities. We also estimated total physical activity and exercise amount per day using a dedicated software. In addition to the JALSPAQ, we asked whether the participants belonged to junior sports teams and whether they participated in club activities in middle and high school.

### 2.3. Statistical Analysis

We divided the participants into two groups (high-risk and low-risk groups) based on the results of the locomotive syndrome risk tests and mean values of women in their 20s (standing on one leg from a stand of 30 cm or less, and the 2-step value is lower than 1.55) [18]. The Japanese Orthopedic Association recommends maintaining an average value for each age group, and this cutoff value is set based on that [17]. We estimated a normal distribution from data on body composition, physical fitness test, walking function, posture, and physical activity. We then compared the mean of the normal distribution data using student’s *t*-test with the non-normal distribution data using the Mann-Whitney’s U test. To clarify which measurement factor increased the locomotive syndrome risk, the objective variable was stratified into two groups by locomotive syndrome risk and the explanatory variables were divided into six items, and multivariate logistic regression analysis (likelihood ratio test using the variable increase method) was performed. The odds ratios and confidence intervals were also calculated. In addition, we used the chi-squared (χ^2^) test to compare the risk level and the presence or absence of membership in junior sports teams or participation/no participation in club activities during middle and high school. IBM SPSS Statistics ver. 22.0 for Windows (IBM Corp., Armonk, NY, USA) was used in this statistical analysis, and the significance level was set at 5% (two-sided test).

### 2.4. Ethical Considerations

This study was carried out with the approval of the Ethics Review Committee of our Women’s University (No. 15–19). The study was conducted in accordance with The Code of Ethics of the World Medical Association (Declaration of Helsinki) for experiments involving humans. Informed consent was obtained from each participant or their legally authorized representative. The measurement data and questionnaires remained anonymous and linked by ID before the statistical analysis.

## 3. Results

### 3.1. Characteristics of the Participants

Table 1 shows the physical, walking, and posture indices of 100 participants. The locomotive syndrome risk tests results showed that the median value of the start-up test was 20 cm, and the average of the two-step values was 1.54 ± 0.12. Of these 100 participants, 43.6% of the participants met the criteria for people in their 20s, and 77.2% met the criteria for people in the 30s. In the normal walking test, the average stride length was 74.3 ± 6.9 cm, and the average speed was 1.5 ± 0.2 m/s. In the maximum walking test, the average stride length was 85.7 ± 12.1 cm, and the average speed was 2.5 ± 0.4 m/s. In addition, in the standing posture test, the forward degree was 2.3 ± 2.0 cm on average. The average front degree was 2.3 ± 2.0 cm in the standing posture test.

### 3.2. Comparison of Results Assessed by Locomotive Syndrome Risk

We divided the participants into two groups based on the locomotive syndrome risk test results and compared the results (Table 2). The low-risk group had significantly lower body fat mass, body fat percentage, and degree of forward lean (anterior degree) than the high-risk group. The low-risk group had a significantly higher maximum walking speed than the high-risk group. More participants in the low-risk group belonged to junior sports teams and participated in club activities during middle and high school than participants in the high-risk group (Table 3).

### 3.3. Relationship between Locomotive Syndrome Risk and Each Measurement Item

We performed a binomial logistic regression analysis to determine which of the measures increased the locomotive syndrome risk (Table 4). Factors for locomotive syndrome risk included body fat mass, maximum walking speed, and anterior degree. Increased body fat mass and forward and decreased maximum walking speed were found to increase the locomotive syndrome risk.

## 4. Discussion

This study evaluated the relationship between the risk of locomotive syndrome in young Japanese women and the standing posture and walking pattern. Our results indicated that the high-risk group had lower speed and stride length of maximum walking than the low-risk group. In addition, the high-risk group had a more prone posture than the low-risk group. In this study, 65.0% of the participants were included in the high-risk group, a very high proportion compared to 26.6% in a previous study using similar cutoff values to Uesugi et al. [12]. The reason may be that this study’s participants were biased toward those with a slightly higher risk. However, we reaffirmed that a certain proportion of young women was at high risk for locomotive syndrome.

At first, regarding the physical characteristics, the low-risk group had a lower body fat mass and lower body fat percentage than the high-risk group. In addition, the results of the logistic regression analysis showed that body fat mass was significantly associated with the locomotive syndrome risk. However, no significant difference was observed in other items regarding body mass, such as BMI and muscle mass. These results were also similar to a previous study’s [12]. The high-risk group is considered to include a high proportion of individuals with “hidden obesity”, i.e., individuals who are not obese but have a low lean mass proportion.

Second, the low-risk group had a significantly higher lower extremity extension power and lower-limb strength-to-weight ratio than the high-risk group regarding motor function. However, no significant difference was found in the two items measuring balance ability between the two groups. The stand-up test conducted in this study measures muscle exertion of one leg’s lower limbs and balance ability [17,18]. Therefore, it is suggested that the locomotive syndrome risk in young people is more associated with lower limb muscle exertion than balance ability. The results of this study for young women are similar to those of these previous studies. Furthermore, the quadriceps muscle cross-sectional area decreases by approximately 40% from the 20s to the 80s [14]. Therefore, we believe that young women, similar to the elderly, need to take measures to prevent muscle mass reduction and muscle weakness.

Third, the low-risk group had a higher maximum walking speed than the high-risk group did, and logistic regression analysis showed a negative correlation between maximum walking speed and locomotive syndrome risk. Many studies have reported that walking speed decreases with age [2,5]. Other reports have shown that locomotive syndrome and sarcopenia in the elderly are strongly associated with diminished gait due to motor dysfunction [24,25]. They have also indicated that age-related loss of walking speed increases the locomotive syndrome risk and sarcopenia. In addition, walking stride length generally reduces the locomotive syndrome risk [26]. This is suggested to maintain lower limb strength, balance, and hip flexibility [27,28]. Although it was the maximum walking in our study, we confirmed that the locomotive syndrome risk in young people was associated with walking speed. Therefore, maintaining the current gait function may help prevent locomotive syndrome. In addition, there was a significant positive correlation between stride length and speed of maximum walking, and lower limb muscle strength [16,29,30]. Many studies have reported that lower limb muscle mass and lower limb muscle strength are closely related to gait function. Hayashida et al. demonstrated in a study with elderly Japanese people that lower limb muscle mass and leg extension were correlated with daily walking speed, and they considered that it was important to prevent the decrease in muscle mass and muscle strength to suppress the deterioration of walking function [31]. Hughes et al. revealed that it was important to prevent the weakness of the lower limb muscles to prevent deterioration of functions of daily living [32].

Fourth, in the standing posture, the anterior degree of posture was significantly higher in the high-risk group than in the low-risk group. The results of the logistic regression analysis also showed a significant association with the locomotive syndrome risk. These results suggest that the high-risk group may have a stooped posture. Fukuda stated that the ideal posture is achieved when the center of gravity of the head, trunk, and lower limbs is located in a straight line when standing, and the head is located in the front, which consequently leads to stiff shoulders and low back pain. Positioning the head in front of the body increases the sway of the center of gravity, which increases fatigue and impairs motor function [33]. Kim et al. also found that patients with sarcopenia have ears and shoulders located more anteriorly than healthy people [34]. Therefore, standing posture may also be a risk factor for locomotive syndrome. It is important to retain an upright posture from a young age.

Fifth, due to the questionnaire, the current physical activity amount and exercise amount were not associated with the locomotive syndrome risk. However, the low-risk group included significantly more individuals who had previously belonged to athletic clubs than the high-risk group. A previous study with female university students reported that middle and high school students belonging to an athletic club had high athletic abilities and were highly aware of their exercise and eating habits [35,36]. Therefore, it was suggested that the amount of past exercise may influence the locomotive syndrome risk among female university students.

There were limitations to this cross-sectional study. The results of this study could not be used to identify causal relationships. In addition, the sample size of this study was relatively small, comprising 100 female university students. This may have created a selection bias during the recruitment stage. Therefore, this result may lack the representativeness of the same generation and more samples will need to be accumulated in the future. Furthermore, as the cutoff value of locomotive syndrome in this study was the average value of young adults, the participants presently experienced no issues with daily activities. Therefore, In the future, it is required to examine whether locomotive syndrome risk in young adults may lead to locomotive syndrome in the elderly.

## 5. Conclusions

We found that a certain number of young women are at locomotive syndrome risk. This risk was related to walking patterns, standing posture, and past exercise habits. In the future, it is required to examine whether locomotive syndrome risk in young adults may lead to locomotive syndrome in the elderly. Young women need the opportunity to be informed, such as in the university, to prevent future long-term care.

## Figures and Tables

**Figure 1 healthcare-08-00565-f001:**
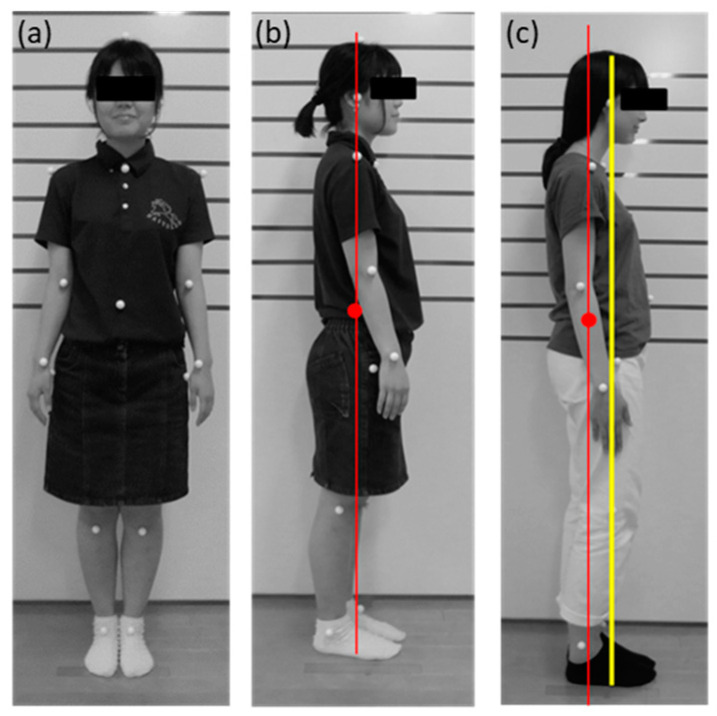
Standing posture. Coronal plane (**a**) and Sagittal plane (ideal (**b**) and forward lean posture (**c**)).

**Table 1 healthcare-08-00565-t001:** Physical measurements ^†^.

Variable		Total (*n* = 100)
Height	(cm)	157.9 ± 5.3
Body mass	(kg)	50.0 ± 5.3
Body mass index	(kg/m^2^)	20.0 ± 1.8
Skeletal muscle mass	(kg)	20.1 ± 2.3
Body fat mass	(kg)	12.6 ± 3.0
Body fat percentage	(%)	25.1 ± 4.4
Calf circumference	(cm)	35.1 ± 5.8
Lower extremity extension power	(Watts)	527.7 ± 194.9
Start-up	(cm)	19.7 ± 11.1
Two-steps value	(m/m)	1.54 ± 0.12
Normal walking stride length	(cm)	74.3 ± 6.9
walking speed	(m/s)	1.5 ± 0.2
Maximum walking stride length	(cm)	85.7 ± 12.1
walking speed	(m/s)	2.5 ± 0.4
Degree of forward lean	(cm)	2.3 ± 2.0

^†^ mean ± standard deviation.

**Table 2 healthcare-08-00565-t002:** Comparison of physical composition, physical fitness, gait function, posture, and physical activity by locomotive syndrome risk group.

Variable		Low-Risk Group	High-Risk Group	*p*-Value
		(*n* = 35)	(*n* = 65)	
Height	(cm) ^†^	156.9 ± 6.2	158.5 ± 4.6	0.135
Body mass	(kg) ^†^	48.9 ± 4.4	50.6 ± 5.6	0.136
Body mass index	(kg/m^2^) ^†^	19.9 ± 1.7	20.1 ± 1.9	0.566
Skeletal muscle mass	(kg) ^†^	20.2 ± 2.1	20.1 ± 1.9	0.909
Body fat mass	(kg) ^†^	11.6 ± 2.2	13.2 ± 4.6	0.004
Body fat percentage	(%) ^†^	23.7 ± 14.9	25.9 ± 3.2	0.016
Calf circumference	(cm) ^‡^	34.3 (33.0–36.0)	34.5 (33.0–36.3)	0.315
Lower extremity extension power	(Watts) ^†^	618.3 ± 182.1	476.4 ± 184.8	<0.001
Lower-limb strength-to-weight ratio	(Watts/kg) ^†^	10.6 ± 3.2	7.9 ± 3.0	<0.001
Normal walking stride length	(m/minute) ^‡^	74.2 (70.8–82.0)	74.2 (70.0–78.8)	0.546
walking speed	(cm) ^‡^	1.6 (1.4–1.7)	1.5 (1.4–1.6)	0.213
Maximum walking stride length	(m/minute) ^‡^	88.8 (83.3–93.3)	86.8 (80.7–90.8)	0.350
walking speed	(cm) ^‡^	2.7 (2.5–2.9)	2.4 (2.1–2.6)	<0.001
Degree of forward lean	(cm) ^‡^	0.8 (0.1–1.7)	2.6 (1.9–4.2)	<0.001
Total amount of physical activity	(METs·h) ^‡^	34.7 (31.9–36.9)	34.3 (31.7–36.7)	0.737
Total amount of exercise	(METs·h) ^‡^	0.0 (0.0–0.7)	0.0 (0.0–0.8)	0.906

^†^ mean ± standard deviation; *t*-test. ^‡^ median (25th–75th percentile); Mann-Whitney U-test.

**Table 3 healthcare-08-00565-t003:** Comparison of participants who took part in club activities during middle or high school by locomotive syndrome risk group ^†^.

Variable		Participated in Club Activities		*p*-Value
		Yes	No	
In middle school	Low-risk group	30 (85.7%)	5 (14.3%)	0.003
	High-risk group	37 (56.9%)	28 (43.1%)	
In high school	Low-risk group	25 (71.4%)	10 (28.6%)	0.002
	High-risk group	25 (38.5%)	40 (61.5%)	

^†^ χ^2^ test.

**Table 4 healthcare-08-00565-t004:** Logistic regression model for the prediction of locomotive syndrome incidence ^†^.

	β_i_	Adjusted Odds Ration	95% CI	*p*-Value
β_0_	3.605			
Body fat mass	0.203	1.225	1.000–1.502	0.050
Maximum walking speed	−2.520	0.080	0.019–0.341	0.001
Degree of forward lean	0.436	1.546	1.164–2.053	0.003
Nagelkerke R^2^	0.420			

^†^ Binomial logistic regression analysis (likelihood ratio test using the variable increase method).

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
