# Peer review of "The Relationship between Locomotive Syndrome Risk, Gait Pattern, and Standing Posture in Young Japanese Women: A Cross-Sectional Study"

_healthcare, 2020, doi:10.3390/healthcare8040565_

Round 1
Reviewer 1 Report
The manuscript entitled “The relationship between locomotive syndrome risk, gait pattern, and standing posture in young Japanese women: a cross-sectional study” by Uesugi and colleagues is devoted to one of the numerous health problems of modern society, even young people appear at risk of locomotive syndrome.
The MS is well written, the research is clearly described. My major concern is about statistics presented in Table 2. Some of the parameters are compared by t-test (presumably Welch's t-test) and the other – by Mann-Whitney U-test. The reasons for one or another criteria should be indicated. Moreover t-tests are applicable only if the data are normally distributed, I couldn’t find a note about the corresponding verification.
The physical units of normal walking stride and walking speed appeared swapped in Table 2.
I also wonder whether the social group of the young female students affect the statistics of the research. Obviously young people have to study more to enter and stay in the university in favour of being physically active. May be in future the authors will consider more uniform pool of participants to have equal numbers of measurements in the groups of low and high risks.
Author Response
Dear Reviewer 1,
Please see the attachment.
Thank you.

Reviewer 2 Report
The main objective of this paper is to identify locomotive syndrome risk in a university sample of women, and to evaluate the relationship of gait and posture with the previously identified risk. The subject is of interest, since the knowledge of the risk factors for motor syndromes are relevant, but there are very important methodological weaknesses, which do not counsel its publication in the present form.
One of the main problems is that the authors did not properly justify and support the objectives and methods. In fact, although the locomotive syndrome has been recently described (doi:10.1007/s00776-007-1202-6.), some references (2-4) used to report the physical deterioration in elderly subjects are too old. Further, there is not enough and/or relevant references of the origin of each measurement and evaluation performed and their metric features in terms of validity and reliability are lacking.
The external validity of the study is, at least, questionable. The sample was exclusively composed of 100 young university women, and it is difficult to know its representativeness. The intrinsic variability of the physical performance of the tests, and the specific characteristics of the university individuals in terms of physical activity, intake, and quality of life prevents a generalization of the results to other stages of life and other young women.
Other concerns:
Introduction:
- Page 1, line 41: “Therefore, stabilizing the standing posture and preventing impaired walking function lead to preventive care.” Please, support with references.
- Page 2, lines 43: Is the references 8 (book) and 9 (published in 2004) the most suitable ones?
- Page 2, lines 44-46: “In addition, it has also been reported that reduced lower limb muscle strength in elementary school students could be problematic [10,11]”. Problematic for what?
- Page 2, lines 52-54: “Therefore, the purpose of this study was to measure the gait index, motor function, and body composition in young women. Additionally, it aimed to clarify the actual state of gait function and posture and assess their relationship with locomotive syndrome risk.” These objectives do not fit with those of the abstract and title. Please, be consistent in the description and determination of the objectives and hypotheses of the study along the whole text.
Material and methods
- Page 2, lines 57-63: Please, detail the inclusion/exclusion criteria.
- Page 2, lines 66-74: There are no data about the recruitment of subjects, the total number of subjects assessed, the causes of exclusion...
Further, more information should be provided to reproduce the sample size estimation.
“All measurements were obtained on a specific day in October 2015.” Who and how performed all the measurements in this specific day?
- Page 3, line 96-106: “We asked the participants to pose for photographs the way they thought they looked the most beautiful.” Why the most beautiful photograph? It is not a scientific criterion.
How the photographs were performed? Was any software used to analyze the images?
“Using that analysis, we calculated the center of gravity (COG). We drew a midline from the COG and measured the distance between the line and ear position.” Please, detail the whole processes. The location of the COG is imprecise.
“The distance was calculated as the "degree of forward lean (cm)" of the head (Fig. 1) [15].” Is reference 15 (book) the most suitable one?
Figure 1: Which are image a)/image b)/image c)?
- Page 4, line 125-126: “Specifically, we changed the wording from “work” to “school life” and included items such as part-time jobs in the leisure activities.” Were these changes validated? Any change in a validated questionnaire should be validated in the target population.
- Page 4, line 126-127: “We also estimated total physical activity and exercise amount per day.” How were these features determined?
- Page 4, line 135: A normal distribution estimation is lacking.
Results
Table 4: Was the R2 determined?
Discussion
- Page 7, line 256-258: “Furthermore, as the cutoff value of locomotive syndrome in this study was the average value of young adults, the participants presently experienced no issues with daily activities.” I agree with the authors. In fact, the relations among the physical tests and the apparition of a locomotive syndrome in the future is too speculative to support the conclusions of the study.
Please, correct the typos along the text.
Author Response
Dear Reviewer 2,
Please see the attachment.
Thank you.

Reviewer 3 Report
The authors studied the relationship between locomotive syndrome risk and walking pattern, standing posture, and past exercise habits in young women between the age of 18 and 23. There were certain correlations observed. The locomotive syndrome has been linked to the risk of falls and fractures in the elderly population. How would symptoms of the locomotive syndrome in the young population linked to the risk of falls and fracture in the future? The significance of the project needs to be better presented, link the observed locomotive syndrome in the tested population to potential clinical risks. What are the risks of the “high-risk” group facing, more falls or fractures in the future?
The tests employed in the study for the locomotive syndrome, the two-step test, the stand-up test, and the 25-question geriatric locomotive function scale, were developed for the elderly populations. Their reliability, sensitivity, and validity have never been studied in the young population investigated in this project, the 18-23 years old. Please justify the use of these tests in this population.
There are few tests that need literature support, see details below. The validity and reliability of these tests need to be reported if the authors developed these tests themselves.
“First” has been mentioned in the abstract and in the discussion (Line 14 and 193). Please note “first” does not justify the value of a project. “First” is only important if you have filled a gap in the literature. It is only important if you have studied something significant. The word “first” here does not signify any significance.
The discussions should not only be related to the difference between the groups, they should also be related to commonly recognized clinical risks. For example, “…the low-risk group had a lower body fat mass and lower body fat percentage than the high-risk group.” (Lines 202-3). This comparison should have been put in the backdrop that the fact that both groups had a perfectly healthy body fat percentage for their age and sex group.
Suggestion for minor edits:
Line 35, “… reduction in stride length is more important …”
Line 38, “… and decrease in arm swing range of motion.”
Line 66, 68, 71, Table 1 & 2, 210: “weight” should have been “mass”; “body weight” should have been “body mass.”
Lie 73, “…thickest part of both thighs was measured…”
Lines 90-94, the walking test needs to be supported with relevant citation.
Line 99, “Using that analysis, we calculated the center of gravity (COG).” More detail needed here and need to be supported by the literature.
Lines 164-165, Table 1, “Lower limb strength” was not mentioned in the method section. Is this the “leg extension force” mentioned in Line 108? Please be consistent with the use of terminology. Its unit should not be “Watts” if force were measured. “Lower leg circumference” mentioned in Tables 1 and 2, was not mentioned in the method section either. Is this the “lower limb circumference” mentioned in Line 72? It should be “thigh circumference” if these are referring to the same measurement. Again, please be consistent with the use of terminology. “Walking stride” should have been “Walking stride length” in both Tables 1 and 2.
Line 175-176, Table 2, the unit for “Degree of forward lean” should not be “(METs・hour).” Then units for stride length, speed, and total amount of exercise were mislabeled.
Author Response
Dear Reviewer 3,
Please see the attachment.
Thank you.

Round 2
Reviewer 3 Report
The authors have addressed all of my concerns.